# Effects of galvanic vestibular stimulation on bodily ownership and postural control: An experimental examination with counterbalanced randomization of stimulus conditions

Kerem Ersin[1]*, Berna Özge Mutlu[2]

**1** Department of Audiology, Faculty of Health Science, Istanbul Medipol University, Istanbul, Turkey,
**2** Department of Audiology, Graduate School of Health Sciences, Istanbul Medipol University, Istanbul, Turkiye

* kersin@medipol.edu.tr

## Abstract

Vestibular stimulation influences both bodily ownership and postural control. Although previous studies in the literature have examined the effects of galvanic vestibular stimulation (GVS) on body ownership and balance separately, their combined and time-dependent effects remain insufficiently explored. This study investigated how GVS modulates multisensory integration over time by assessing bodily ownership and postural control within the same participants. A within-participant design was used with four conditions: Baseline (pre-GVS), Sham (60-min post-GVS placebo), 30-min post-GVS, and 60-min post-GVS. Forty-eight healthy adults completed all conditions. Balance performance was assessed via the Single-Leg Stance (SLS) and Fukuda Stepping Test (FST), while bodily ownership was measured using the Rubber Hand Illusion (RHI) questionnaire. Balance performance on the SLS showed a significant reduction at 30 minutes post-GVS, with values returning toward baseline by 60 minutes. In contrast, angular deviation on the FST decreased significantly at both 30 and 60 minutes post-GVS. RHI ownership scores increased at both post-stimulation time points, with the most pronounced increase observed at 60 minutes. The sham condition also elicited increases in RHI scores, indicating possible expectancy-related effects. Overall, these findings indicate time-dependent and task-specific effects of GVS on bodily ownership and postural control. The results are consistent with adaptive sensory reweighting processes that differentially affect static and dynamic balance measures. Further research in clinical populations and using longer stimulation protocols is required to determine the extent to which these short-term effects translate into sustained functional benefits.

**Data availability statement:** All relevant data are contained within the manuscript.

**Funding:** The publication fee is to be supported by Istanbul Medipol University as per the institutional Scientific Activities Incentive Directive for Q1 category journal. No additional external funding was received for this study. The funders had no role in study design, data collection and analysis, decision to publish, or preparation of the manuscript.

**Competing interests:** The authors have declared that no competing interests exist.

## Introduction

Human awareness of the body and its surroundings arise from multisensory integration. Inputs from the visual, auditory, tactile, proprioceptive and vestibular systems are combined in the central nervous system to support both bodily perception and postural control [1]. Through this integration, individuals locate the body in space, regulate interactions with the external environment and maintain balance [2]. The vestibular system encodes head movement and position, interacts with other sensory modalities, and contributes to perceptual and motor functions [3]. Disruption of these processes is frequently encountered in otolaryngology practice, highlighting the relevance of understanding how vestibular inputs contribute to body representation and postural regulation.

Bodily ownership captures a specific facet of multisensory integration. The Rubber Hand Illusion (RHI) is a well-established experimental model used to investigate this phenomenon. By manipulating cross-modal synchrony and sensory conflict, it reveals how the brain integrates visual, tactile and proprioceptive information into body representation [4]. Its experimental precision makes it a useful bridge between basic mechanisms and clinical questions.

Emerging evidence indicates that vestibular inputs can influence these multisensory processes. Both electrical and caloric vestibular stimulation change the weighting of visual and proprioceptive cues and influence both ownership measures and postural responses [5,6]. Vestibular function is also central to static and dynamic postural stability [7–9]. However, most studies have examined body ownership and balance separately. Evidence directly addressing their concurrent modulation by vestibular stimulation within the same individuals remains limited.

The present study addresses this gap by examining the time-dependent effects of galvanic vestibular stimulation on bodily ownership and postural control using a within-participant design. By assessing RHI outcomes alongside static and dynamic balance measures at multiple post-stimulation time points, this approach aims to provide an integrated perspective on how vestibular input contributes to multisensory reweighting across perceptual and motor domains.

## Materials and methods

### Participants

This study was conducted between December 1 and December 31, 2019, at the Audiology and Vestibular Laboratories of İstanbul Medipol University. A total of forty-eight healthy volunteers aged between 18 and 30 years (mean age = 24.1 years, SD = 2.7) were recruited.

All participants were right-handed and reported normal or corrected-to-normal vision and hearing. Individuals were excluded if they had any contraindication to vestibular stimulation, a self-reported history of vestibular, neurological, or psychiatric disorders, current use of medications known to affect the vestibular or auditory systems, or severe visual or auditory impairment.

The study protocol was reviewed and approved by the İstanbul Medipol University Non-Invasive Clinical Research Ethics Committee (Decision No. 1023, dated 27 November 2019). Written informed consent was obtained from all participants prior to enrollment.

## Sample size

An a priori power analysis with G*Power 3.1 for a within-subjects repeated-measures analysis with four conditions was used. The significance level (α).05 and power was .80. A medium effect size (f = 0.25) was assumed, together with an expected correlation among repeated measures of 0.50 and a sphericity correction factor of 0.75. Based on these assumptions, a total sample size of forty-eight participants was required.

## Design and conditions

A within-participant repeated-measures design was employed, comprising four experimental conditions: Baseline (pre-GVS), Sham (control, 60-minute post-GVS placebo), 30-minute post-GVS, and 60-minute post-GVS. Every participant completed all four conditions.

The order of the three stimulation sessions (Sham, 30-minute GVS, and 60-minute GVS) was counterbalanced across participants to minimize potential order and practice effects. The Baseline session was scheduled first for all participants to establish a standardized pre-intervention reference and to avoid carryover effects from vestibular stimulation. Each session was separated by a minimum washout period of 48 hours.

Outcome measures were collected once during each condition. For the stimulation sessions, assessments were conducted immediately following completion of the session. A study flow diagram illustrating the sequence of conditions is presented in Fig 1.

The primary outcome was postural control as assessed by balance performance. The secondary outcome was bodily ownership as measured by the RHI ownership score.

## Randomization

The order of the three stimulation sessions (Sham control, 30-minute GVS, and 60-minute GVS) was randomized across participants. Randomization sequences were generated using a computer-based permuted block procedure. A study coordinator maintained the allocation sequence and disclosed the assigned condition only to the device operator at the time of session setup.

The Baseline condition was conducted as a separate no-stimulation session and was scheduled first for all participants in order to standardize pre-intervention measurements. Participants were informed that they would complete one Sham session and two active stimulation sessions but were not informed of the order of these sessions.

Outcome assessors were blinded to the stimulation condition. Sessions were separated by a minimum interval of 48 hours to reduce potential carryover effects. The allocation process was monitored to ensure that participant numbers were approximately balanced across the different session orders.

## Galvanic vestibular stimulation

Galvanic vestibular stimulation (GVS) was delivered using the Modius Sleep device (Neurovalens Ltd., Belfast, UK). This device was selected based on previously published sham-controlled studies demonstrating its safety and feasibility in human participants [7]. Disposable gel electrodes (3.2 cm diameter) were placed bilaterally over the mastoid processes on alcohol-cleansed skin, with impedance maintained below 5 kΩ.

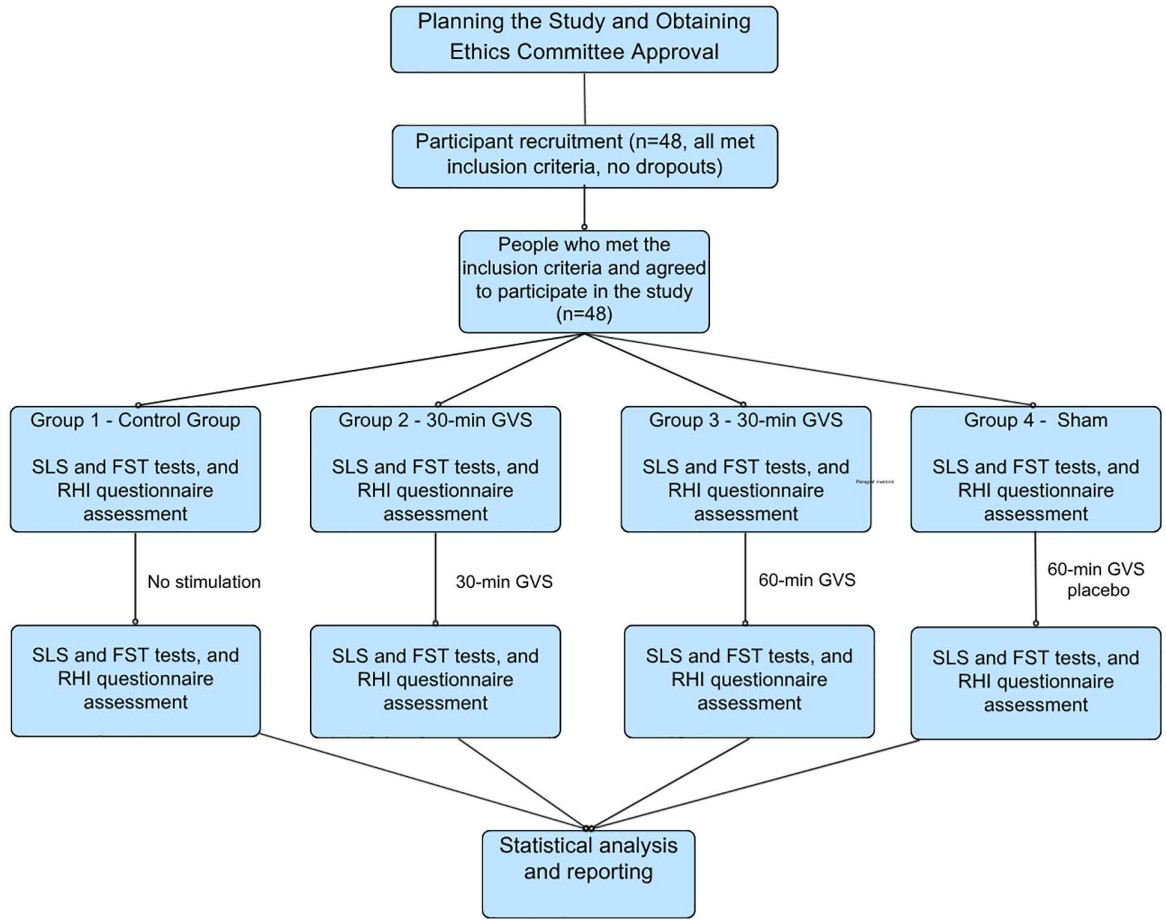

**Fig 1. Schematic Representation of The Study Design.**

Active stimulation administered as binaural-bipolar, sub-Hz, charge-balanced alternating current with a peak amplitude of 0.5 mA and a 2-second ramp-up and ramp-down period [7]. Stimulation durations was 30 minutes in the GVS-30 condition and 60 minutes in the GVS-60 condition. The estimated current density was approximately 0.06 mA/cm²

For the sham condition, the headset was applied in an identical manner, and a brief ramp-up phase was delivered to reproduce initial cutaneous sensations, after which the stimulation output was reduced to zero while device indicators remained active.

Safety was monitored during and immediately after each session using a predefined checklist covering dizziness, nausea, headache, and skin irritation. Any moderate or severe adverse symptoms prompted immediate termination of the session and clinical evaluation according to predefined stopping criteria.

## Balance tests

The Fukuda Stepping Test was performed with participants standing barefoot, with eyes closed and arms extended forward. Participants were instructed to march in place for 50 steps at a pace of 1 Hz guided by a metronome. Angular deviation (degrees) and forward and lateral displacement (centimeters) were recorded using floor-mounted angular and distance markings (Fig 2a).

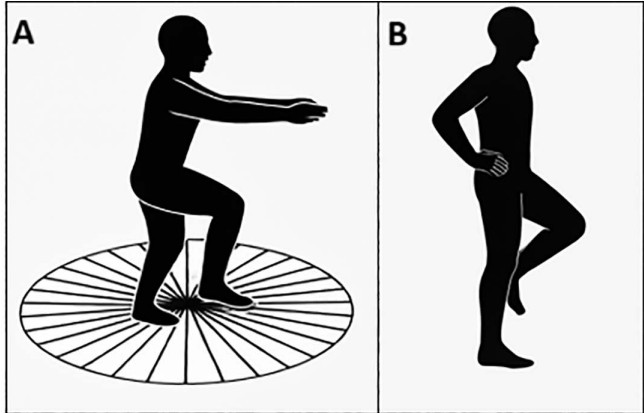

**Fig 2. Balance performance assessments. (A)** Single-Leg Stance (SLS) test, assessing static balance with eyes closed, and **(B)** Fukuda Stepping Test (FST), assessing postural orientation and angular deviation during stepping with eyes closed.

Single-Leg Stance (SLS) performance was assessed barefoot with eyes closed and arms resting alongside the body. Participants were asked to maintain balance on both the dominant and non-dominant legs for a maximum duration of 30 seconds. Two trials were performed for each leg within each condition, and the mean value was used for analysis (Fig 2b). The number of trials was limited to reduce fatigue and cumulative task effects across repeated experimental sessions.

## Rubber hand illusion

RHI paradigm was employed as an established experimental model of multisensory body ownership, allowing assessment of how vestibular stimulation modulates bodily self-perception through multisensory integration [4].

During the RHI procedure, the participant's right hand was positioned behind an opaque screen, while a life-sized rubber hand was placed in a plausible anatomical alignment approximately 15 cm from the participant's real hand. Visual access to the participant's own hand was completely occluded.

In the experimental condition, synchronous visuotactile stimulation was applied for three minutes by brushing the participant's concealed hand and the visible rubber hand simultaneously at a frequency of 1 Hz using identical brushes and movement trajectories. In the control condition, asynchronous brushing was applied for three minutes using the same materials and timing, but with a temporal offset between the tactile and visual inputs (Fig 3). This asynchronous condition served as an internal control for the RHI paradigm.

Following each condition, participants completed a nine-item self-report questionnaire assessing subjective ownership experiences. Items were rated on a Likert scale ranging from −3 (strongly disagree) to +3 (strongly agree). The questionnaire items are presented in Table 1. The validated Turkish version of the RHI questionnaire was administered to all participants [4,10].

## Blinding

Participants received a standardized briefing about the study procedures before enrollment. They were informed that they would complete three stimulation sessions consisting of one Sham session and two active GVS sessions for different durations, but they were not informed about the order of these sessions. At the beginning of each session, participants were unaware of whether the stimulation delivered would be active or sham session order was assigned using a computer-generated randomization sequence and was disclosed only to the device operator at the time of session setup.

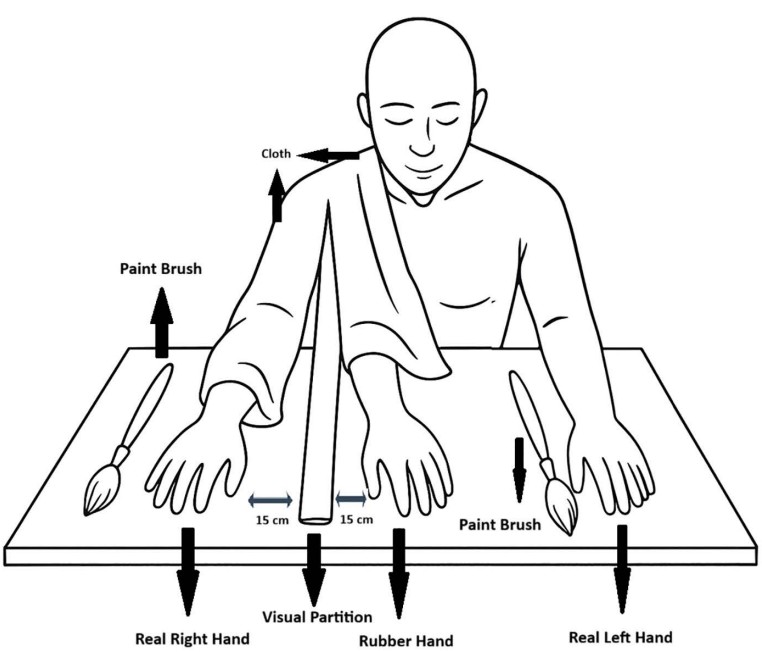

**Fig 3. Rubber Hand Illusion procedure.**

**Table 1. Rubber hand illusion questionnaire.**

| 1. | I felt as if the rubber hand was part of my body. |
|---|---|
| 2. | I felt as if the rubber hand was my hand. |
| 3. | I felt as if the touch was caused by the brush touching the rubber hand. |
| 4. | I felt the touch of the brush in the location where I saw the rubber hand being touched. |
| 5. | I felt as if my real hand was turning rubbery. |
| 6. | I felt as if I had no longer a right hand, as if my right hand had disappeared. |
| 7. | It seemed as if the touch I was feeling came from somewhere between my own right hand and the rubber hand. |
| 8. | It seems as if I had more than one right hand. |

Outcome assessors remained blinded to the stimulation condition throughout data collection. The device operator could not be blinded because the operator was required to initiate the stimulation. However, the operator did not participate in outcome assessment or data scoring. All procedures were conducted according to a standardized operating protocol to minimize potential non-verbal or procedural bias.

Following each session, participants were asked to indicate whether they believed the stimulation had been active or Sham and to rate their confidence in this judgment. These responses were used as a descriptive check of blinding success rather than as a formal assessment of blinding validity.

## Safety monitoring

Adverse events were monitored during and immediately after each session using a predefined checklist. The checklist included symptoms of dizziness, nausea, headache, and skin irritation. Any adverse symptom rated by the participant

as moderate or severe prompted immediate termination of the session and clinical assessment according to predefined stopping criteria.

### Statistical analysis

Statistical analyses were performed using SPSS version 22.0 (IBM, Armonk, NY, USA). The distributional properties of the outcome variables were examined using the Shapiro–Wilk test. Several key outcome variables showed significant deviations from normality, particularly balance and delta measures, as detailed in Supplementary S1 Table. Therefore, nonparametric statistical methods were applied.

Differences across the four experimental conditions (pre-GVS, Sham control, 30-minute post-GVS, and 60-minute post-GVS) were assessed using the Friedman test. When a significant overall effect was detected, pre-specified pairwise comparisons were conducted using Wilcoxon signed-rank tests. Comparisons of interest included pre-GVS versus 30-minute post-GVS, pre-GVS versus 60-minute post-GVS, Sham versus 30-minute post-GVS, and Sham versus 60-minute post-GVS. To control for multiple comparisons, Holm–Bonferroni correction was applied, and adjusted p-values were evaluated with an overall significance threshold of $p < .05$.

In addition, exploratory correlation analyses were performed on delta scores calculated as the difference between post-condition and baseline values. These analyses were conducted to examine associations between changes in bodily ownership, measured by the Rubber Hand Illusion ownership score ($\Delta$RHI), and changes in postural outcomes, including Single-Leg Stance performance for the left and right legs ($\Delta$SLS-left and $\Delta$SLS-right) and Fukuda Stepping Test angular deviation ($\Delta$FST). Spearman's rank correlation coefficients were used as a nonparametric measure of association for each condition (30-minute post-GVS, 60-minute post-GVS, and Sham).

## Results

### Participant flow and adverse events

All 48 participants completed the four conditions (baseline, sham, 30-min post-GVS, 60-min post-GVS), and no dropouts or missing data occurred. No moderate or severe adverse events were reported. Mild transient dizziness (n = 2) was observed, but all resolved spontaneously without intervention.

### Single-leg stance test

SLS time differed significantly across the four conditions for both right and left stance ($p < .001$; Table 2; Fig 4). Post hoc pairwise comparisons revealed that SLS times were significantly reduced at 30 min post-GVS compared with pre-GVS for both the left and right stance ($p < .01$). By contrast, no significant differences were observed between pre-GVS and sham conditions for either stance ($p > .05$).

Comparisons between 30 min and 60 min post-GVS showed a significant increase in SLS times, indicating a recovery toward baseline performance by 60 min post-stimulation (both $p < .001$). No significant differences were found between pre-GVS and 60 min post-GVS or between sham and 60 min post-GVS for either stance ($p > .05$), suggesting that the reduction in static balance was transient and specific to the 30-min post-GVS condition.

### Fukuda stepping test

Fukuda stepping angles differed significantly across experimental conditions ($p < .001$; Table 2; Fig 4). Relative to pre-GVS, angular deviation was significantly reduced at both 30 min and 60 min post-GVS ($p < .001$), indicating improved dynamic orientation following vestibular stimulation.

**Table 2. Results of SLS, FST, and RHI assessments in baseline, sham, 30-min, and 60-min GVS conditions.**

| Conditions | RHI Questionnaire Score Median (IQR) | SLS (Left) Score Median (IQR) | SLS (Right) Score Median (IQR) | FST Angle Score Median (IQR) |
|---|---|---|---|---|
| Pre-GVS | −2.0 (3.0) | 11.0 (5.0) | 12.0 (5.0) | 42.0 (31.0) |
| 30 min post-GVS | 5.0 (4.5) | 6.0 (8.5) | 6.0 (8.0) | 25.0 (32.5) |
| 60 min post-GVS | 7.0 (3.5) | 12.0 (3.5) | 13.0 (3.5) | 24.0 (8.5) |
| Sham | 6.0 (3.0) | 12.0 (4.0) | 13.0 (4.0) | 34.0 (15.0) |
| Intergroup Comparison p value | <.001* | <.001* | <.001* | <.001* |
| Pre-GVS vs 30 min post-GVS p value | <.001* | .002* | 001* | <.001* |
| Pre-GVS vs 60 min post-GVS p value | <.001* | .003* | <.001* | <.001* |
| Pre-GVS vs Sham p value | <.001* | .043* | .202 | <.001* |
| 30 min post-GVS vs 60 min post-GVS p value | .006* | <.001* | <.001* | .987 |
| 30 min post-GVS vs Sham p value | .712* | <.001* | <.001* | .105 |
| 60 min post-GVS vs Sham P value | .004* | .269 | .682 | <.001* |

*$p < 0.05$; IQR: Interquartile Range; SLS: Single-Leg Stance; FST: Fukuda Stepping Test; RHI: Rubber Hand Illusion

The sham condition also showed a significant reduction in angular deviation compared with pre-GVS ($p < .001$). Pairwise comparisons demonstrated no significant difference between sham and 30 min post-GVS ($p > .05$), whereas angular deviation at 60 min post-GVS was significantly lower than in the sham condition ($p < .001$). These findings indicate a progressive improvement in dynamic postural orientation, with the most pronounced effect observed at 60 min post-GVS.

### Rubber hand illusion questionnaire

RHI ownership scores differed significantly across the four conditions ($p < .001$; Table 2; Fig 4). Compared with pre-GVS, RHI scores were significantly higher at both 30 min and 60 min post-GVS ($p < .001$), indicating increased bodily ownership following vestibular stimulation.

The sham condition also yielded significantly higher RHI scores than pre-GVS ($p < .001$). Pairwise comparisons showed no significant difference between 30 min post-GVS and sham ($p \geq .05$). In contrast, RHI scores at 60 min post-GVS were significantly higher than those observed under sham stimulation ($p < .01$), suggesting a time-dependent enhancement of bodily ownership beyond expectancy effects.

### Exploratory correlation analyses

Exploratory Spearman correlation analyses of change scores (Δ values) at 30 min post-GVS revealed significant associations between changes in bodily ownership and changes in postural control (Table 3). Increases in ΔRHI scores were moderately and negatively correlated with changes in SLS performance for both the left and right stance ($\rho = -0.477$, $p < .001$). In addition, a weaker but significant negative correlation was observed between ΔRHI scores and changes in Fukuda stepping angles ($\rho = -0.324$, $p = .030$).

These findings indicate a short-term coupling between subjective bodily ownership and objective postural measures following vestibular stimulation.

At 60 min post-GVS, exploratory correlation analyses revealed no significant associations between ΔRHI scores and changes in SLS or Fukuda stepping angles (all $p \geq .05$; Table 4). Although balance-related change scores remained correlated with each other, bodily ownership changes were no longer associated with postural outcomes at this later point.

Under sham stimulation, no significant correlations were observed between ΔRHI scores and any postural change measures ($p > .05$; Table 5). Strong correlations were present among balance-related variables themselves, but no

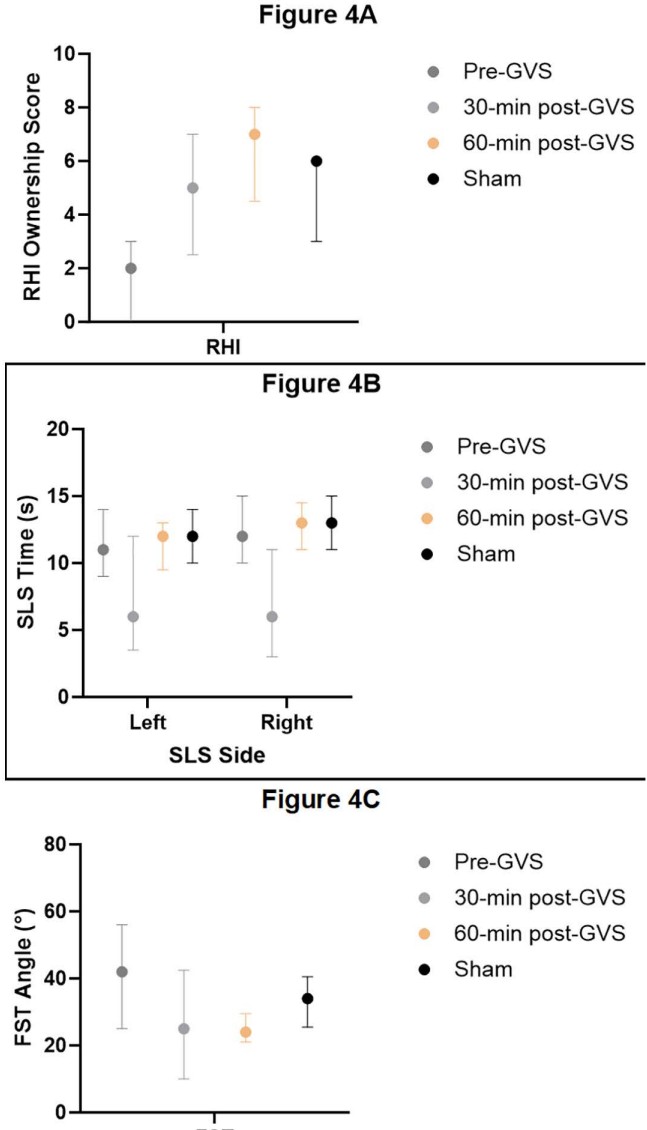

**Fig 4. Median and interquartile ranges of (A) Rubber Hand Illusion (RHI) ownership scores, (B) Single-Leg Stance (SLS) time for left and right stance, and (C) Fukuda Stepping Test (FST) angular deviation across experimental conditions (pre-GVS, 30-min post-GVS, 60-min post-GVS, and sham).** Points represent medians and error bars indicate interquartile ranges.

evidence of coupling between bodily ownership and postural control was detected in the absence of active vestibular stimulation.

## Discussion

Prior works show that GVS reliably alters human balance and self-motion perception. Electrical vestibular stimulation modulates postural sway patterns and subjective sensations of rotation [11,12]. GVS can improve balance metrics, especially in older adults and in patients with vestibular hypofunction [13–15]. Evidence also indicates that vestibular input interacts with visual and somatosensory cues to readjust multisensory weights in a context-dependent manner [16]. In

**Table 3. Exploratory Spearman correlations between change scores at 30 min post-GVS.**

| Variable | ΔSLS–Left | ΔSLS–Right | ΔFST Angle | ΔRHI Score |
|---|---|---|---|---|
| ΔSLS–Left | — | ρ = 0.971<br>p < 0.001* | ρ = 0.701<br>p < 0.001* | ρ = −0.477<br>p < 0.001* |
| ΔSLS–Right | ρ = 0.971<br>p < 0.001* | — | ρ = 0.772<br>p < 0.001* | ρ = −0.477<br>p < 0.001* |
| ΔFST Angle | ρ = 0.701<br>p < 0.001* | ρ = 0.772<br>p < 0.001* | — | ρ = −0.324<br>p = 0.030* |
| ΔRHI Score | ρ = −0.477<br>p < 0.001* | ρ = −0.477<br>p < 0.001* | ρ = −0.324<br>p = 0.030* | — |

Correlation coefficients represent Spearman's rho (ρ). *p < .05. Δ = delta score (post – baseline); RHI = Rubber Hand Illusion ownership score; FST = Fukuda Stepping Test; SLS = Single-Leg Stance.

**Table 4. Exploratory Spearman correlations between change scores at 60 min post-GVS.**

| Variable | ΔSLS–Left | ΔSLS–Right | ΔFST Angle | ΔRHI Score |
|---|---|---|---|---|
| ΔSLS–Left | — | ρ = 0.690<br>p < 0.001* | ρ = 0.041<br>p = 0.782 | ρ = −0.101<br>p = 0.508 |
| ΔSLS–Right | ρ = 0.690<br>p < 0.001* | — | ρ = 0.302<br>p = 0.037* | ρ = −0.170<br>p = 0.265 |
| ΔFST Angle | ρ = 0.041<br>p = 0.782 | ρ = 0.302<br>p = 0.037* | — | ρ = −0.221<br>p = 0.144 |
| ΔRHI Score | ρ = −0.101<br>p = 0.508 | ρ = −0.170<br>p = 0.265 | ρ = −0.221<br>p = 0.144 | — |

Correlation coefficients represent Spearman's rho (ρ). *p < .05. Δ = delta score (post – baseline); RHI = Rubber Hand Illusion ownership score; FST = Fukuda Stepping Test; SLS = Single-Leg Stance.

**Table 5. Exploratory Spearman correlations between change scores under sham stimulation.**

| Variable | ΔSLS–Left | ΔSLS–Right | ΔFST Angle | ΔRHI Score |
|---|---|---|---|---|
| ΔSLS–Left | — | ρ = 0.963<br>p < 0.001* | ρ = 0.448<br>p = 0.001* | ρ = −0.243<br>p = 0.108 |
| ΔSLS–Right | ρ = 0.963<br>p < 0.001* | — | ρ = 0.406<br>p = 0.004* | ρ = −0.235<br>p = 0.120 |
| ΔFST Angle | ρ = 0.448<br>p = 0.001* | ρ = 0.406<br>p = 0.004* | — | ρ = −0.131<br>p = 0.391 |
| ΔRHI Score | ρ = −0.243<br>p = 0.108 | ρ = −0.235<br>p = 0.120 | ρ = −0.131<br>p = 0.391 | — |

Correlation coefficients represent Spearman's rho (ρ). *p < .05. Δ = delta score (post – baseline); RHI = Rubber Hand Illusion ownership score; FST = Fukuda Stepping Test; SLS = Single-Leg Stance.

parallel, the bodily ownership literature shows that vestibular signals can selectively influence visuo-tactile integration and the feeling of ownership [17,18].

This study adds two contributions. First, we assessed vestibular stimulation effects within the same participants using both postural control measures (SLS and the FST) and a bodily ownership assay (RHI). Second, we tracked effects over time by comparing four conditions: baseline, sham, 30 minutes post-stimulation, and 60 minutes post-stimulation. This design allowed a more integrated interpretation of the relation between perceptual representations and motor outputs at successive time points. Although prior work has examined GVS effects on bodily ownership, concurrent postural

measurements with multi-time-point follow-up are uncommon [17,18]. Our protocol therefore provides a framework for examining how vestibular input may differentially influence multisensory body representations and postural behavior over time.

The SLS time showed a significant reduction at 30 minutes post-GVS, with values returning toward baseline by 60 minutes. This pattern suggests that brief vestibular stimulation may transiently disrupt static postural control, with subsequent normalization over time. One possible explanation is a temporary sensory mismatch induced by vestibular perturbation, rather than a sustained impairment of balance processing. The observed recovery is consistent with previously described rapid sensory reweighting processes within vestibular–cerebellar networks [2,9,12,19]. No side-to-side differences were observed when dominant and non-dominant legs were analyzed separately. Both legs showed a similar transient decrement at 30 minutes followed by recovery by 60 minutes. Together, these findings indicate a short-lived effect of GVS on static balance that resolves within approximately one hour.

The reduction of rotation angle on the FST observed at 30 minutes post-GVS, with a further decrease at 60 minutes, indicates a time-dependent modulation of dynamic orientation control. The initial attenuation at 30 minutes may reflect non-specific factors such as task familiarity, expectancy, or practice-related learning, as a comparable improvement was also observed in the sham condition. These effects suggest that early changes in angular deviation are not exclusively attributable to vestibular stimulation. By contrast, the more pronounced reduction in rotation angle at 60 minutes, which differed significantly from both baseline and sham, supports the presence of a stimulation-specific contribution. This delayed effect is consistent with the engagement of adaptive multisensory processes that integrate vestibular input with visual and proprioceptive cues to stabilize directional control. Rather than indicating a single localized mechanism, these findings align with models of distributed vestibular–cerebellar involvement in dynamic balance regulation, whereby directional biases are gradually reduced through experience-dependent reweighting [2,16,19]. Similar improvements in directional accuracy during dynamic balance tasks have been reported following repeated exposure to vestibular input in prior studies [13].

By contrast, the increase in RHI scores observed at 60 minutes was significantly greater than both pre-GVS and sham conditions, pointing to a more stable alteration in sensory weighting. This delayed effect is consistent with previous research indicating that prolonged vestibular stimulation can induce stronger changes in multisensory processing [16,20]. Rather than reflecting an immediate recalibration, the findings suggest a gradual strengthening of bodily ownership that emerges with sustained vestibular input. Together, these results support the view that vestibular signals contribute to the reweighting of bodily self-experience over time, particularly when stimulation is maintained for longer durations. In line with these findings, exploratory correlation analyses indicated that greater increases in RHI scores tended to be associated with smaller angular deviations on the FST, whereas no consistent associations were observed with SLS performance. Although these relationships did not remain statistically significant after correction for multiple comparisons, their overall direction is compatible with the possibility that prolonged GVS may influence dynamic orientation more strongly than static balance measures.

Rather than indicating a direct coupling between ownership and postural control, these correlations should be interpreted as hypothesis-generating. They align with previous accounts suggesting that extended vestibular stimulation may gradually bias multisensory integration toward visual–vestibular recalibration in tasks requiring directional control [16,19]. In the sham condition, higher RHI scores showed a weak tendency to co-occur with reduced SLS performance. While this association was not statistically robust, it may reflect transient changes in postural strategy related to expectancy or attentional factors rather than a genuine alteration of balance capacity. Such interpretations are consistent with evidence that cognitive context and anticipation can modulate sensory weighting and short-term balance regulation [16,19,21].

Paromov et al. (2025) reported that GVS enhanced static postural control in patients with vestibular hypofunction, with the largest improvement observed under conditions of increased sensory challenge, such as eyes closed on a foam surface [20]. Although the present study was conducted in healthy participants, a partially comparable pattern emerged in the FST, where angular deviation decreased following GVS and was most pronounced at 60 min post-stimulation. A

modest improvement was also observed in the sham condition, suggesting that early changes may partly reflect practice-related learning or expectancy effects. However, the significantly greater reduction in deviation observed at 60 min post-GVS compared with sham supports the presence of stimulation-specific effect beyond nonspecific learning. These findings are consistent with earlier work by Carmona et al. (2011), who demonstrated that vestibular stimulation could bias bodily self-representation under multisensory conflict, emphasizing the role of vestibular input in recalibrating perceptual reference frames rather than uniformly enhancing performance [22]. In the present study, this recalibration appears to extend from perceptual domains to motor behavior over time. In contrast, SLS performance showed a transient decline at 30 min with recovery toward baseline by 60 min, indicating that static and dynamic balance measures responded differently to vestibular perturbation. Taken together, the combined patterns observed across the FST, SLS, and RHI suggest that vestibular input modulates postural control and bodily ownership in a context- and task-dependent manner, likely through adaptive sensory reweighting rather than uniform enhancement across perceptual and motor domains.

In multisensory balance control, vestibular contributions are modulated by the relative reliability of visual and somatosensory cues. When visual information is stable and coherent, greater reliance on vision is typically observed, whereas during visual conflict or deprivation, vestibular and somatosensory inputs tend to be upweighted [16]. Within this framework, the concurrent increase in RHI ownership scores between 30 and 60 minutes and the reduction in angular deviation on the FST are compatible with a shift in sensory weighting over time. Rather than implying a direct alteration of vestibular gain, these findings suggest a gradual adjustment in how visual and vestibular information are integrated during tasks requiring dynamic orientation. Previous studies have shown that changes in visual reliability, ground conditions, and repeated exposure to sensory conflict can modify postural strategies and balance behavior [23–25]. Accordingly, the co-occurrence of enhanced bodily ownership and improved directional control in the present study is consistent with adaptive sensory reweighting processes, whereby vestibular input contributes to the recalibration of multisensory representations in a context-dependent manner.

## Conclusion

This study demonstrates that GVS is associated with time-dependent changes in bodily ownership and postural control. Short-duration stimulation was accompanied by a transient disruption of static balance, whereas longer stimulation durations were linked to improvements in dynamic orientation and increases in bodily ownership. Together, these findings are consistent with the notion that vestibular input contributes to adaptive sensory reweighting across perceptual and motor domains.

Although these results were obtained in healthy young adults and reflect short-term effects, they highlight mechanisms that may be relevant for future investigations of vestibular-based interventions. Further studies in clinical populations and using longer stimulation protocols are needed to determine whether these time-dependent effects can be translated into sustained therapeutic benefits.

## Limitations

This study has several limitations. First, the sample consisted of healthy young adults, which restricts generalizability of the findings to clinical populations. Second, only two SLS trials per leg were conducted, which may provide lower test–retest reliability compared with protocols using three or more trials. Third, bodily ownership was assessed using subjective questionnaire measures, without inclusion of objective proprioceptive drift. Fourth, the study focused on short-term effects of GVS, and therefore did not address longer-term adaptation, habituation, or potential aftereffects. Finally, although a sham condition was included, expectancy-related influences cannot be entirely ruled out, particularly for perceptual measures.

## Methodological considerations for future research

Future studies would benefit from greater standardization of GVS protocols to reduce between-study heterogeneity. Key parameters, including current intensity, waveform characteristics, electrode size, and electrode placement, should be reported in a consistent and transparent manner to facilitate comparison across investigations. Where feasible, individualized intensity calibration may enhance tolerability and ecological validity. In the assessment of balance outcomes, the inclusion of instrumented posturography is recommended, with separate reporting of anteroposterior and mediolateral center-of-pressure components to better capture sensory reweighting processes. Similarly, investigations of bodily ownership would be strengthened by combining subjective self-report measures with objective indices, such as proprioceptive drift or neurophysiological markers. Finally, longer-duration stimulation protocols or repeated GVS sessions should be explored to evaluate adaptation, habituation, and potential carryover effects. Such approaches may help clarify whether the time-dependent effects observed in healthy individuals extend to sustained changes and whether similar mechanisms operate in clinical populations.

## Supporting information

**S1 Table. Normality test results.** This table provides the Shapiro–Wilk test statistics and associated p-values for the RHI scores, SLS times, and FST angular deviations across all four conditions (Baseline, Sham, 30-min GVS, and 60-min GVS).
(DOCX)

**S2 Raw Data. Minimal underlying dataset.** This spreadsheet contains the fully anonymized raw data, including individual Likert scale responses for the RHI questionnaire and trial-by-trial performance metrics for all balance assessments.
(XLSX)

## Acknowledgments

The authors gratefully acknowledge the contribution of the students who participated in this study.

## Author contributions

**Conceptualization:** Kerem Ersin, Berna Özge Mutlu.

**Formal analysis:** Kerem Ersin.

**Investigation:** Kerem Ersin, Berna Özge Mutlu.

**Methodology:** Kerem Ersin, Berna Özge Mutlu.

**Resources:** Berna Özge Mutlu.

**Supervision:** Berna Özge Mutlu.

**Writing – original draft:** Kerem Ersin.

**Writing – review & editing:** Kerem Ersin, Berna Özge Mutlu.

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
