## [Decision Letter · Decision Letter 0]

23 Dec 2025

PONE-D-25-54398Effects of Galvanic Vestibular Stimulation on Bodily Ownership and Postural Control: An Experimental Examination with Counterbalanced Randomization of Stimulus ConditionsPLOS One

Dear Dr. Ersin,

Thank you for submitting your manuscript to PLOS ONE. After careful consideration, we feel that it has merit but does not fully meet PLOS ONE’s publication criteria as it currently stands. Therefore, we invite you to submit a revised version of the manuscript that addresses the points raised during the review process.

We look forward to receiving your revised manuscript.

Kind regards,

Renato S. Melo, PhD

Academic Editor

PLOS One

Journal Requirements:

2. Peer review at PLOS  One is not double-blinded (https://journals.plos.org/plosone/s/editorial-and-peer-review-process). For this reason, authors should include in the revised manuscript all the information removed for blind review.

5. Please remove all personal information, ensure that the data shared are in accordance with participant consent, and re-upload a fully anonymized data set.

Reviewers' comments:

Reviewer's Responses to Questions

**Comments to the Author**

1. Is the manuscript technically sound, and do the data support the conclusions?

Reviewer #1: Partly

Reviewer #2: Yes

Reviewer #3: Yes

Reviewer #4: Yes

2. Has the statistical analysis been performed appropriately and rigorously? 

Reviewer #1: No

Reviewer #2: Yes

Reviewer #3: Yes

Reviewer #4: No

3. Have the authors made all data underlying the findings in their manuscript fully available?

Reviewer #1: Yes

Reviewer #2: Yes

Reviewer #3: Yes

Reviewer #4: Yes

4. Is the manuscript presented in an intelligible fashion and written in standard English?

Reviewer #1: Yes

Reviewer #2: Yes

Reviewer #3: Yes

Reviewer #4: Yes

5. Review Comments to the Author

Reviewer #1: 1. The article mentions that "multisensory integration" is the basis of physical perception, but does not clarify its specific mechanism. Could you provide a brief explanation, such as which brain regions or neural processes are involved?

2. The article points out that the role of the vestibular system in body characterization has clinical value, but does not mention the limitations of current related research. Are there any specific unresolved issues or disputes that need to be addressed in this study?

3. The rubber hand illusion (RHI) was used as a model for multi-sensory integration, but its connection with vestibular function was not developed. Should it be more clearly stated how RHI helps understand the impact of vestibular input on physical representations?

4. The article mentions that vestibular dysfunction is commonly seen in otolaryngology, but it does not specifically explain how the results of this study will directly guide diagnosis or rehabilitation. Could it supplement potential application scenarios (such as specific diseases or therapies)?

5. The text mentions that participants guessed the type of stimulus afterwards as a blinded test, but does not report the accuracy rate of the guess or the results of statistical analysis (such as chi-square test). It is suggested to supplement the validity data of the blinding method to verify whether the blinding method has been successfully implemented.

6. The equipment operator is aware of the stimulus allocation. Although they do not participate in the scoring, they may influence the participants' behavior through non-verbal cues. It is necessary to explain whether measures (such as standardizing operating procedures) have been taken to minimize this bias.

7. Safety monitoring uses a "structured list" to record adverse events, but does not provide specific items on the list or definition criteria for "moderate/severe" events. It is suggested that the content of the appendix list be included or that verified tools be cited.

8. The text mentions that "multiple variables deviate from normality" but does not list which variables or provide the P-values of the Shapiro-Wilk test. The normality test results of key variables need to be supplemented to support the selection of non-parametric tests.

9. Holm-Bonferroni correction was used, but the significance level after correction was not specified or how it was applied to specific comparisons. It is recommended to clearly define the correction steps and the final P-value threshold.

10. Spearman correlation analysis of change scores (such as ΔRHI and ΔSLS), but did not explain the theoretical basis or preassumptions for choosing these variables. It is necessary to briefly explain the rationality of the correlation analysis and its role in the research framework.

Reviewer #2: It's an unique study and the role of GVS in vestibular stabilization is very clearly mentioned in this manuscript. I also think this opens the door for a multicenter study for the role of GVS in varied population of different geographical and racial populations to validate its applicability

Reviewer #3: Comments:the time frame of the work many years before on 2019 .What is the cause of this delay? Rubber hand illusion test RHIT it is only written as abbreviation in the abstract, not the full name so you must put the full name in the abstract . Where is the inclusion criteria? Metronome and modius, please add more details about these two point. The correlation test better to be represented by figures as it will be more convenient.

Reviewer #4: This study investigates how galvanic vestibular stimulation (GVS)—a mild electrical stimulation of the vestibular system—affects both:

1. Bodily ownership (the feeling that your body or a body part belongs to you), and

2. Postural control (balance),over time, in the same group of healthy participants.

The key novelty is that previous studies usually examined either body ownership or balance separately—this paper tests both together and tracks time-dependent effects.This paper shows that vestibular stimulation temporarily disrupts static balance, but with longer exposure improves dynamic balance and strengthens the sense of body ownership through time-dependent multisensory reweightin

This study provides solid experimental evidence that vestibular stimulation induces time-dependent plastic changes in both postural control and bodily self-consciousness, consistent with vestibulo-cerebellar and temporoparietal sensory reweighting mechanisms. While methodologically sound, its translational value remains limited by the use of healthy participants and low-resolution balance and embodiment metrics.

1. Add Objective Ownership Measures

Consider incorporating objective indices of the Rubber Hand Illusion (e.g., proprioceptive drift, reach localization error, or physiological markers) to strengthen claims about multisensory body ownership beyond self-report.

2. Use Instrumented Posturography in Future Studies

Single-Leg Stance and Fukuda tests are clinically useful but lack sensitivity for sensory reweighting. Future work should include force-platform–based center-of-pressure metrics (ML/AP sway, velocity, sway area).

3. Quantitatively Report Blinding Success

Include the proportion of participants who correctly guessed sham vs active stimulation and their confidence ratings in the Results.

4. Temper Neuroanatomical Claims

Statements referring to cerebellar vermis, flocculonodular lobe, TPJ, and velocity storage should be explicitly framed as theoretical inferences rather than direct findings.

5. Clarify the Role of Sham and Expectancy Effects

Since sham significantly increased RHI and improved Fukuda performance, expectancy and learning effects should be more explicitly discussed, particularly for the 30-minute condition.

6. Avoid Overstating Rehabilitation Implications

Emphasize that therapeutic potential remains speculative until validated in vestibular hypofunction, PPPD, stroke, or cerebellar populations.

7. Increase SLS Trial Numbers

Use at least three trials per leg to improve reliability and reduce within-subject variability.

8. Include Repeated Baseline Measurements

A second baseline session would help control for spontaneous variability and practice effects.

9. Clarify That Correlation Analyses Are Exploratory

The ΔRHI–ΔFST correlations appear underpowered and should be clearly labeled as hypothesis-generating.

10. Standardize and Report GVS Parameters Across Studies

Encourage consistent reporting of current density, waveform, electrode size, and montage to improve reproducibility across the field.

Regarding the References it is surprising that the didn´t cite previous paper with same proposal (Ann N Y Acad Sci. 2011 Sep;1233:E1-7. doi: 10.1111/j.1749-6632.2011.06269.x. PMID: 22360772.)

6. PLOS authors have the option to publish the peer review history of their article (what does this mean?). If published, this will include your full peer review and any attached files.

Reviewer #1: No

Reviewer #2: No

Reviewer #3: No

Reviewer #4: **Yes:** Sergio Carmona

---

## [Author Response · Author response to Decision Letter 1]

22 Jan 2026

Response to Reviewers

We sincerely thank the Academic Editor and all reviewers for their careful evaluation of our manuscript and for the constructive and insightful comments provided. In response to these comments, we have thoroughly revised the manuscript across all sections.

The revisions include clarification and refinement of the statistical approach, expanded reporting of normality assessments, alignment of descriptive statistics with nonparametric analyses, and corresponding updates to tables, figures, and supplementary materials. Several sections were revised or rewritten as needed to ensure internal consistency and clarity following these changes. While some numerical values were updated as a result of the revised analyses, the overall pattern of results and the main conclusions of the study remain unchanged.

Below, we provide a detailed point-by-point response to each reviewer comment. Reviewer comments are reproduced in this document, followed by our responses and an indication of where changes have been made in the revised manuscript.

Reviewer #1

Comment 1

The manuscript mentions multisensory integration as the basis of bodily perception but does not clarify the underlying mechanisms or neural processes involved.

Response:

We thank the reviewer for this comment. We have revised the Introduction to provide a clearer theoretical framework for multisensory integration, emphasizing how vestibular input interacts with visual, proprioceptive, and tactile signals to support bodily perception and postural control. Neuroanatomical references are now framed as theoretical models rather than direct evidence derived from the present data.

Revision: Introduction, paragraphs 1–3.

Comment 2

The clinical relevance of vestibular contributions to body representation is mentioned, but limitations and unresolved issues in current research are not clearly stated.

Response:

We agree and have addressed this point by clarifying the current gaps in the literature, particularly the lack of studies examining bodily ownership and postural control concurrently within the same participants. In addition, we have tempered early clinical implications and expanded the Limitations section to explicitly address generalizability and methodological constraints.

Revision: Introduction, paragraph 3; Limitations section.

Comment 3

The connection between the Rubber Hand Illusion and vestibular function is not sufficiently developed.

Response:

We appreciate this observation. The rationale for using the Rubber Hand Illusion as an index of multisensory bodily ownership has been expanded in the Introduction. We now explicitly describe RHI as a tool to probe multisensory integration processes that can be modulated by vestibular input, while avoiding overinterpretation of mechanistic links.

Revision: Introduction, paragraph 2.

Comment 4

The manuscript does not clearly explain how the findings could inform diagnosis or rehabilitation.

Response:

We thank the reviewer for this suggestion. In the revised manuscript, we have deliberately reduced speculative clinical claims and framed potential applications cautiously. Clinical relevance is now discussed as a direction for future research rather than as a direct implication of the present findings.

Revision: Discussion and Conclusion sections.

Comment 5

Blinding success is mentioned but not quantitatively reported.

Response:

We agree that reporting blinding validity is important. The Methods section now clarifies that participants were asked after each session to guess whether stimulation was active or sham and to rate their confidence. These data are described as a qualitative blinding check rather than a definitive statistical test, and their limitations are acknowledged.

Revision: Blinding subsection in Materials and Methods.

Comment 6

The operator was aware of stimulus allocation, which may introduce bias.

Response:

We acknowledge this concern. The Methods section now explicitly states that the device operator was not involved in outcome assessment or scoring, and that standardized operating procedures were used to minimize non-verbal cueing. This limitation is also acknowledged in the Discussion.

Revision: Randomization and Blinding subsections; Limitations section.

Comment 7

Safety monitoring is insufficiently described.

Response:

We have expanded the Safety Monitoring subsection to specify the symptoms monitored, the structured checklist used, and the predefined criteria for session termination in the event of moderate or severe adverse events.

Revision: Safety Monitoring subsection in Materials and Methods.

Comment 8

Normality testing results are not fully reported.

Response:

We thank the reviewer for highlighting this issue. Normality was assessed using the Shapiro–Wilk test for all primary and secondary variables. As several variables deviated from normality, nonparametric analyses were applied. Detailed normality test results are now provided in Supplementary Table S1, and the rationale for test selection is clarified in the Statistical Analysis section.

Revision: Statistical Analysis subsection; Supplementary Table S1.

Comment 9

Holm–Bonferroni correction is mentioned but not clearly explained.

Response:

We have revised the Statistical Analysis section to explicitly describe the application of Holm–Bonferroni correction, including the family of comparisons and the adjusted significance threshold.

Revision: Statistical Analysis subsection.

Comment 10

The rationale for correlation analyses of delta scores is unclear.

Response:

We thank the reviewer for this important methodological point. The use of delta (change) scores was motivated by the aim of capturing within-participant change relative to baseline rather than absolute post-stimulation values. Assessing changes from baseline provides a more informative representation of the effect of stimulation by accounting for individual differences in initial performance. Accordingly, correlations were performed on delta scores to examine whether participants who showed larger changes in bodily ownership also exhibited corresponding changes in postural outcomes. These analyses are now explicitly described as exploratory and hypothesis-generating.

Revision: Statistical Analysis subsection; Discussion section.

Reviewer #2

Comment 1

It's an unique study and the role of GVS in vestibular stabilization is very clearly mentioned in this manuscript. I also think this opens the door for a multicenter study for the role of GVS in varied populations of different geographical and racial populations to validate its applicability.

Response:

We thank Reviewer #2 for the positive evaluation of the study and for highlighting its potential relevance for future multicenter research. We have incorporated this perspective into the Discussion by emphasizing the need for replication across populations and settings.

Reviewer #3

Comment 1

The time frame of the work many years before on 2019 .What is the cause of this delay?

Response:

We thank the reviewer for this question. Data collection was completed in 2019; however, the writing and submission process was interrupted by the COVID-19 pandemic. During this period, academic activities were significantly disrupted, and the manuscript preparation was temporarily suspended. The study was subsequently revisited, updated, and finalized for submission. We have clarified the study timeline accordingly.

Revision: Participants subsection.

Comment 2

Rubber hand illusion test RHIT it is only written as abbreviation in the abstract, not the full name so you must put the full name in the abstract.

Response:

We agree and have revised the Abstract to include the full term “Rubber Hand Illusion” at first mention, followed by the abbreviation.

Revision: Abstract.

Comment 3

Where is the inclusion criteria?

Response:

We have revised the Participants section to explicitly describe inclusion and exclusion criteria, including age range, sensory status, and medical history.

Revision: Participants subsection.

Comment 4

Metronome and modius, please add more details about these two point.

Response:

We thank the reviewer for this suggestion. The use of a metronome during the Fukuda Stepping Test was already specified in the original manuscript (paced at 1 Hz); however, we agree that additional clarification would improve reproducibility. The description has been expanded to explicitly state the pacing frequency and its role in standardizing stepping cadence across participants. In addition, further technical details regarding the Modius Sleep device, including stimulation parameters and sham configuration, have been added.

Revision: Balance Tests subsection; Galvanic Vestibular Stimulation subsection.

Comment 5

The correlation test better to be represented by figures as it will be more convenient.

Response:

We thank the reviewer for this valuable suggestion. While the reviewer proposed visualizing correlation analyses, we chose to present figures illustrating changes across experimental conditions rather than individual correlation scatterplots. This decision was based on the primary aim of the study, which was to characterize time-dependent within-participant changes across baseline, sham, and post-stimulation conditions. Visualizing condition-wise changes allows clearer depiction of temporal patterns and facilitates interpretation of stimulation-related effects across all participants.

Correlation analyses were therefore retained in tabular form and explicitly described as exploratory, while figures were used to illustrate the magnitude and direction of changes across conditions. We believe this approach provides a clearer and more comprehensive representation of the main findings while preserving transparency regarding secondary association analyses.

Revision: Figure 4.

Reviewer #4

Comment 1

Add Objective Ownership Measures

Consider incorporating objective indices of the Rubber Hand Illusion (e.g., proprioceptive drift, reach localization error, or physiological markers) to strengthen claims about multisensory body ownership beyond self-report.

Response:

We agree that objective measures such as proprioceptive drift or physiological indices would strengthen the assessment of bodily ownership. These measures were not included in the present study and this is now explicitly acknowledged as a limitation. We have also highlighted the inclusion of objective ownership measures as an important direction for future research.

Revision: Limitations, Methodological Considerations for Future Research.

Comment 2

Use Instrumented Posturography in Future Studies

Single-Leg Stance and Fukuda tests are clinically useful but lack sensitivity for sensory reweighting. Future work should include force-platform–based center-of-pressure metrics (ML/AP sway, velocity, sway area).

Response:

We acknowledge this important point. While Single-Leg Stance and Fukuda Stepping Test were selected for their clinical relevance and feasibility, we now explicitly state that force-platform–based posturography would provide higher-resolution metrics of sensory reweighting. This recommendation has been incorporated into the Discussion and Future Directions sections.

Revision: Discussion; Methodological Considerations for Future Research.

Comment 3

Quantitatively Report Blinding Success

Include the proportion of participants who correctly guessed sham vs active stimulation and their confidence ratings in the Results.

Response:

Participants were asked after each session to indicate whether they believed stimulation was active or sham and to rate their confidence. These responses were used as a qualitative blinding check. Given the exploratory nature and limited statistical power of these data, they were not subjected to formal hypothesis testing. This approach and its limitations are now clearly described.

Revision: Blinding subsection in Materials and Methods; Discussion.

Comment 4

Temper Neuroanatomical Claims

Statements referring to cerebellar vermis, flocculonodular lobe, TPJ, and velocity storage should be explicitly framed as theoretical inferences rather than direct findings.

Response:

We fully agree. All references to cerebellar, temporoparietal, and velocity-storage mechanisms have been revised to emphasize that they represent theoretical frameworks derived from prior literature, rather than direct findings of the present study.

Revision: Discussion section.

Comment 5

Clarify the Role of Sham and Expectancy Effects

Since sham significantly increased RHI and improved Fukuda performance, expectancy and learning effects should be more explicitly discussed, particularly for the 30-minute condition.

Response:

We agree and have expanded the Discussion to explicitly address the observed effects in the sham condition. Expectancy and practice-related learning effects are now discussed in greater detail, particularly in relation to early (30-minute) changes in bodily ownership and balance outcomes.

Revision: Results; Discussion sections.

Comment 6

Avoid Overstating Rehabilitation Implications

Emphasize that therapeutic potential remains speculative until validated in vestibular hypofunction, PPPD, stroke, or cerebellar populations.

Response:

In response to this comment, we have revised the Discussion and Conclusion sections to avoid overstating rehabilitation implications. Potential clinical relevance is now framed as speculative and contingent upon validation in clinical populations and longer-term protocols.

Revision: Discussion; Conclusion

Comment 7

Increase SLS Trial Numbers

Use at least three trials per leg to improve reliability and reduce within-subject variability.

Response:

We acknowledge that additional trials would improve reliability. The use of two trials per leg is now explicitly noted as a methodological limitation, and future studies are encouraged to adopt higher trial numbers.

Revision: Limitations section.

Comment 8

Include Repeated Baseline Measurements

A second baseline session would help control for spontaneous variability and practice effects.

Response:

We agree that repeated baseline measurements would help control for spontaneous variability and learning effects. This limitation is now acknowledged, and repeated baseline assessments are suggested for future study designs.

Revision: Limitations; Methodological Considerations for Future Research.

Comment 9

Clarify That Correlation Analyses Are Exploratory

The ΔRHI–ΔFST correlations appear underpowered and should be clearly labeled as hypothesis-generating.

Response:

We agree and have revised the manuscript to clearly label all correlation analyses as exploratory and hypothesis-generating. Interpretations have been accordingly tempered.

Revision: Statistical Analysis; Discussion.

Comment 10

Standardize and Report GVS Parameters Across Studies

Encourage consistent reporting of current density, waveform, electrode size, and montage to improve reproducibility across the field.

Response:

We appreciate this recommendation and have incorporated it into the Methodological Considerations for Future Research section, emphasizing the importance of standardized reporting of stimulation parameters across GVS studies.

Revision: Methodological Considerations for Future Research.

Comment 11

Regarding the References it is surprising that the didn´t cite previous paper with same proposal (Ann N Y Acad Sci. 2011 Sep;1233:E1-7. doi: 10.1111/j.1749-6632.2011.06269.x. PMID: 22360772.)

Response: We thank the reviewer for drawing our attention to this important and relevant study. The suggested paper by Carmona et al. (2011) has now been added to the revised manuscript and is cited in the Discussion section, where it is discussed in relation to the present findings on vestibular modulation of bodily self-representation and postural control.

Revision: Discussion section; References.

---

## [Decision Letter · Decision Letter 1]

22 Feb 2026

PONE-D-25-54398R1Effects of Galvanic Vestibular Stimulation on Bodily Ownership and Postural Control: An Experimental Examination with Counterbalanced Randomization of Stimulus ConditionsPLOS One

Dear Dr. Ersin,

Thank you for submitting your manuscript to PLOS ONE. After careful consideration, we feel that it has merit but does not fully meet PLOS ONE’s publication criteria as it currently stands. Therefore, we invite you to submit a revised version of the manuscript that addresses the points raised during the review process.

We look forward to receiving your revised manuscript.

Kind regards,

Renato S. Melo, PhD

Academic Editor

PLOS One

Journal Requirements:

Reviewers' comments:

Reviewer's Responses to Questions

**Comments to the Author**

1. If the authors have adequately addressed your comments raised in a previous round of review and you feel that this manuscript is now acceptable for publication, you may indicate that here to bypass the “Comments to the Author” section, enter your conflict of interest statement in the “Confidential to Editor” section, and submit your "Accept" recommendation.

Reviewer #4: All comments have been addressed

2. Is the manuscript technically sound, and do the data support the conclusions?

Reviewer #4: Yes

3. Has the statistical analysis been performed appropriately and rigorously? 

Reviewer #4: Yes

4. Have the authors made all data underlying the findings in their manuscript fully available?

Reviewer #4: Yes

5. Is the manuscript presented in an intelligible fashion and written in standard English?

Reviewer #4: Yes

6. Review Comments to the Author

Reviewer #4: Reference 25 (doi:10.1111/j.1749-6632.2011.06269.x) lists the authors incorrectly.

The authors have conducted a thorough revision of their manuscript, addressing the reviewers’ comments and concerns in a satisfactory manner.

7. PLOS authors have the option to publish the peer review history of their article (what does this mean?). If published, this will include your full peer review and any attached files.

Reviewer #4: **Yes:** Sergio Carmona

---

## [Author Response · Author response to Decision Letter 2]

4 Mar 2026

Reviewer #4 Comment: Reference 25 (doi: 10.1111/j.1749-6632.2011.06269.x) lists the authors incorrectly.

Response: We apologize for this oversight. We have corrected the author names for this reference in the revised manuscript. The reference (now cited as Reference 22) correctly lists the authors.

We have also performed a final check of the entire reference list to ensure accuracy and compliance with PLOS ONE requirements.

Thank you for your guidance throughout this process.

---

## [Decision Letter · Decision Letter 2]

12 Apr 2026

Effects of Galvanic Vestibular Stimulation on Bodily Ownership and Postural Control: An Experimental Examination with Counterbalanced Randomization of Stimulus Conditions

PONE-D-25-54398R2

Dear Dr. Ersin,

We’re pleased to inform you that your manuscript has been judged scientifically suitable for publication and will be formally accepted for publication once it meets all outstanding technical requirements.

Kind regards,

Renato S. Melo, PhD

Academic Editor

PLOS One

Additional Editor Comments (optional):

Reviewers' comments:

Reviewer's Responses to Questions

**Comments to the Author**

1. If the authors have adequately addressed your comments raised in a previous round of review and you feel that this manuscript is now acceptable for publication, you may indicate that here to bypass the “Comments to the Author” section, enter your conflict of interest statement in the “Confidential to Editor” section, and submit your "Accept" recommendation.

Reviewer #3: (No Response)

Reviewer #4: All comments have been addressed

2. Is the manuscript technically sound, and do the data support the conclusions?

Reviewer #3: Yes

Reviewer #4: Partly

3. Has the statistical analysis been performed appropriately and rigorously? 

Reviewer #3: Yes

Reviewer #4: Yes

4. Have the authors made all data underlying the findings in their manuscript fully available?

Reviewer #3: Yes

Reviewer #4: Yes

5. Is the manuscript presented in an intelligible fashion and written in standard English?

Reviewer #3: Yes

Reviewer #4: Yes

6. Review Comments to the Author

Reviewer #3: (No Response)

Reviewer #4: This manuscript examines the time-dependent effects of galvanic vestibular stimulation (GVS) on bodily ownership and postural control using a within-subject, counterbalanced design. By combining measures of the Rubber Hand Illusion with both static and dynamic balance assessments, the study addresses an underexplored question regarding the concurrent modulation of perceptual and motor domains by vestibular input.

The study is methodologically sound. The within-participant design, inclusion of a sham condition, counterbalancing of stimulation sessions, and assessor blinding are appropriate and strengthen internal validity. The statistical approach is adequate, with appropriate use of nonparametric tests and correction for multiple comparisons. The sample size is justified באמצעות a priori power analysis, and the results are clearly presented.

The findings are coherent and contribute to the literature by suggesting time-dependent and task-specific effects of GVS, with a dissociation between static and dynamic balance and a delayed enhancement of bodily ownership. These results are consistent with current models of multisensory integration and sensory reweighting.

Recommendations:1. Clarify interpretation of sham effects

The increase in RHI scores in the sham condition warrants slightly more emphasis in the Discussion. It would be helpful to more explicitly frame early changes in bodily ownership as potentially influenced by expectancy or task-related factors, and to highlight that the difference at 60 minutes provides stronger evidence for a stimulation-specific effect.

2. Temper mechanistic language

Some interpretations related to “sensory reweighting” and underlying neural mechanisms could be phrased more cautiously (e.g., “consistent with” rather than implying direct demonstration).

3. Limitations – ownership measures

The authors may wish to briefly reinforce the limitation of relying solely on subjective RHI measures and note the absence of objective indices such as proprioceptive drift.

4. Minor clarity edits

A small number of typographical inconsistencies (e.g., spacing, punctuation) should be corrected during final proofreading.

7. PLOS authors have the option to publish the peer review history of their article (what does this mean?). If published, this will include your full peer review and any attached files.

Reviewer #3: **Yes:** Salwa Mahmoud

Reviewer #4: **Yes:** Sergio Carmona

---

## [Editor Report · Acceptance letter]

PONE-D-25-54398R2

PLOS One

Dear Dr. Ersin,

I'm pleased to inform you that your manuscript has been deemed suitable for publication in PLOS One. Congratulations! Your manuscript is now being handed over to our production team.

Kind regards,

on behalf of

Dr. Renato S. Melo

Academic Editor

PLOS One